# A Web-Based Decision Support System for Aircraft Dispatch and Maintenance

Hemmo Koornneef [1,*] , Wim J. C. Verhagen [2] and Richard Curran [1]

1 Faculty of Aerospace Engineering, Delft University of Technology, Kluyverweg 1, 2629HS Delft, The Netherlands; R.Curran@tudelft.nl
2 Department of Aerospace Engineering and Aviation, RMIT University School of Engineering, Building 57, Level 3, 115 Queensberry St., Carlton, VIC 3053, Australia; wim.verhagen@rmit.edu.au
* Correspondence: H.Koornneef@tudelft.nl

**Abstract:** Aircraft dispatch involves determining the optimal dispatch option when an aircraft experiences an unexpected failure. Currently, maintenance technicians at the apron have limited access to support information and finding the right information in extensive maintenance manuals is a time-consuming task, often leading to technically induced delays. This paper introduces a novel web-based prototype decision support system to aid technicians during aircraft dispatch decision-making and subsequent maintenance execution. A system architecture for real-time dispatch decision support is established and implemented. The developed system is evaluated through a case study in an operational environment by licensed maintenance technicians. The system fully automates information retrieval from multiple data sources, performs alternative identification and evaluation for a given fault message, and provides the technician with on-site access to relevant information, including the related maintenance tasks. The case study indicates a potential time saving of up to 98% per dispatch decision. Moreover, it enables digitalization of the—currently mostly paper-based—dispatch decision process, thereby reducing logistics and paper waste. The prototype is the first to provide operational decision support in the aircraft maintenance domain and addresses the lack of correlation between theory and practice often found in decision support systems research by providing a representative case study. The developed custom parser for SGML-based documents enables efficient identification and extraction of relevant information, vastly contributing to the overall reduction of the decision time.

**Keywords:** decision support; aircraft maintenance; dispatch assessment; information retrieval; data integration; mobile tools

## 1. Introduction

Maintenance is an essential element for ensuring the continuing airworthiness of aircraft and, by extension, the primary consideration in aviation: safety. Aircraft only add value to the airline when they perform revenue flights and airlines aim to minimize the so-called aircraft on ground (AOG) time. Traditional scheduled maintenance is based on calendar days, flight hours, or flight cycles, and decades of research on the optimization of maintenance scheduling to maximize aircraft availability has resulted in a solid body of literature (e.g., [1–4]). In recent years, a shift towards data-driven maintenance approaches can be observed, where, instead of a predefined interval, the current health or expected failure of the part or system is the determining factor [5–7]. These approaches aim to minimize the remaining useful life (RUL) of parts and systems before performing maintenance, to reduce maintenance costs and AOG time.

The aforementioned maintenance strategies aim to prevent unexpected failures and their associated costs. Despite these efforts, unscheduled maintenance as a result of unexpected failures remains a relevant factor in airline operations: it is estimated that 5.8% of all flights in Europe experience delays due to unexpected aircraft technical failures, resulting

in EUR 2.8 billion additional costs every year [8]. The actual impact is likely to be even worse, because cancelled flights are registered as having no delay [9]. The cost of delay is modelled in several publications, which all found similar trends in delay cost factors (e.g., maintenance costs, flight and cabin crew salaries and expenses, airport charges, and passenger compensations) for both the EU and US markets [9–14].

### 1.1. Aircraft Dispatch

When an aircraft experiences an unexpected failure, a technician has to assess the situation and determine the best course of action—the so-called dispatch assessment. During this assessment, technicians have to troubleshoot the problem, identify dispatch options, and execute the seemingly best alternative, preferably without causing a delay. One of the key issues that the technicians face is finding task-specific information in maintenance manuals. Aircraft maintenance manuals are very extensive, containing thousands of individual tasks that also cross-reference each other. Finding all relevant information to perform a maintenance task is a tedious process and, as a result, technicians spend as much as 30% of their valuable time searching for information [15]. With the current paper-based way of working, maintenance manuals are not available at the apron (the apron is a defined area, on a land aerodrome, intended to accommodate aircraft for purposes of loading or unloading passengers, mail or cargo, fuelling, parking, or maintenance [16]) and technicians spend additional time driving to the office, to search and print task-specific information and then drive back to the aircraft at the apron to perform the maintenance. A typical turnaround for a narrow-body aircraft requires approximately 50 min according to the manufacturer [17], but airlines aim to minimize the turnaround time. For example, low-cost carriers usually schedule only 30 min for a turnaround. The limited turnaround time in combination with the time required to troubleshoot the problem, acquire relevant information, and perform the maintenance task sometimes leads to considerable time pressure amongst technicians, which is one of many human factors that cause maintenance error [18–20].

### 1.2. Decision Support for Aircraft Maintenance

In contrast to the vast amount of available literature for planned maintenance (i.e., considering strategic and tactical decision-making), research in operational decision-making for reactive aircraft maintenance is scarce. Papakostas et al. [21] address short-term maintenance planning within a multi-criteria decision support framework but focus on the planning of pending maintenance tasks that have been identified previously. Moreover, the authors stress that the framework is highly beneficial for health-monitored components in particular. Dhanisetty et al. [22] developed an approach for multi-criteria decision-making for operational maintenance decisions with time horizons of hours or a few days at maximum, which exceeds the time available during turnaround. Moreover, the decision alternatives are generic (e.g., temporary fix, permanent fix), instead of actual maintenance tasks as described in manuals. To the best of the authors' knowledge, no literature covering decision support for immediate actions when an aircraft experiences an unexpected failure has yet been published.

Looking at decision support systems from a broader perspective, the literature indicates that the main issue in the domain is a lack of correlation between the theoretical systems and the actual system use [23,24] and experts stress that more case studies should be performed to address this issue [25–30].

### 1.3. Research Contribution, Aim, and Scope

From a theoretical perspective, this research contributes to the domain of aircraft maintenance by introducing a novel web-based approach for real-time decision support and subsequent maintenance execution, addressing the difficulties that technicians face with the access to, and usability of, maintenance information, especially at the apron. The approach facilitates the straightforward combination of multiple heterogeneous information sources

and is the first to address operational decision-making for unexpected failures in the aircraft maintenance domain. A custom-built parser enables the automated identification and retrieval of maintenance information that is relevant to a given fault message. Moreover, it addresses the indicated lack of correlation between theory and practice in decision support systems research by end-user evaluation through a case study. While applied in the aircraft maintenance domain, the introduced approach could also be advantageous in other domains with comparable timescales, pressures, and constraints as acting in aviation, leading to similar operational decision-making issues. Examples include offshore assets [31], the nuclear industry [32], and healthcare applications [33].

From an applied perspective, the research aims to resolve the aforementioned issues that technicians face during aircraft dispatch assessment by developing a prototype real-time decision support system that automates the information retrieval process, including alternative identification and evaluation, based on a given fault message. An overview of the alternatives and their consequence is provided to the technician on-site, at the aircraft, through a mobile application and includes direct access to task-specific maintenance documents for support during maintenance execution.

## 2. System Architecture and Key Functions

To address the current issues that technicians face, we introduce a prototype decision support system to aid technicians during aircraft dispatch decision-making and maintenance execution at the apron, and thereby aim to reduce operational disruptions caused by unexpected failures.

### 2.1. Prototype Requirements

Functional requirements specify what the prototype must be able to do from an end-user perspective. The goal is to minimize technically induced flight disruptions by reducing the time that technicians currently need to acquire decision support information, including task-support information for subsequent maintenance execution. This is achieved by automating the information retrieval process and providing the information to the technician independent of the location. Hence, the following functional requirements (FR) are determined:

- **FR1** Provide technicians with a concise overview of the available dispatch options, following on from a fault message, including their expected outcome with respect to the flight schedule;
- **FR2** Provide technicians with access to task-support information;
- **FR3** Provide technicians with the most recent information to ensure safety compliance (i.e., work according to the latest manual revision) and accuracy (i.e., use the latest information for alternative ranking);
- **FR4** Provide technicians with fast (i.e., <1 min) access to relevant maintenance information anywhere (assuming that a wireless network connection (mobile/WiFi) is available);
- **FR5** Provide external applications with access to maintenance documentation.

To achieve the functional requirements, several system requirements (SR) are established. The system should:

- **SR1** automate the information retrieval from maintenance documentation;
- **SR2** split maintenance manuals on task level and transform the source data to a clear visual format;
- **SR3** allow technicians to search maintenance manuals on task level;
- **SR4** be able to integrate and process information from heterogeneous, distributed information sources, given that a wide variety of data formats and sources are used in airline and maintenance operations;
- **SR5** combine information from different sources and generate a concise overview to support dispatch decision-making;

- **SR6** update when new information becomes available (e.g., a new manual revision or defect);
- **SR7** support ubiquitous computing to provide technicians with decision support information independent of their location;
- **SR8** allow connections from external applications to access maintenance task documentation, in order to provide a seamless workflow between applications.

The next sections detail the system architecture (Section 2.2) and the key system functions (Section 2.3) that have been developed to meet these requirements.

### 2.2. System Implementation and Architecture

The web-based decision support prototype is developed in Node.js because it is well-equipped for multi-platform and real-time applications. Node.js is open source and provides high performance, has a rich ecosystem, and is full-stack JavaScript (i.e., both the server and client are programmed in JavaScript). With these characteristics, Node.js is a very suitable platform with respect to system requirements **SR4**, **SR7**, and **SR8**. Combined with HTML and CSS, a prototype with a dynamic user interface has been developed that is accessible through web browsers and an Android app. The system architecture of the prototype for dispatch decision support is shown in Figure 1, having three main elements that will be discussed in the following sections: the server, the client, and the external database.

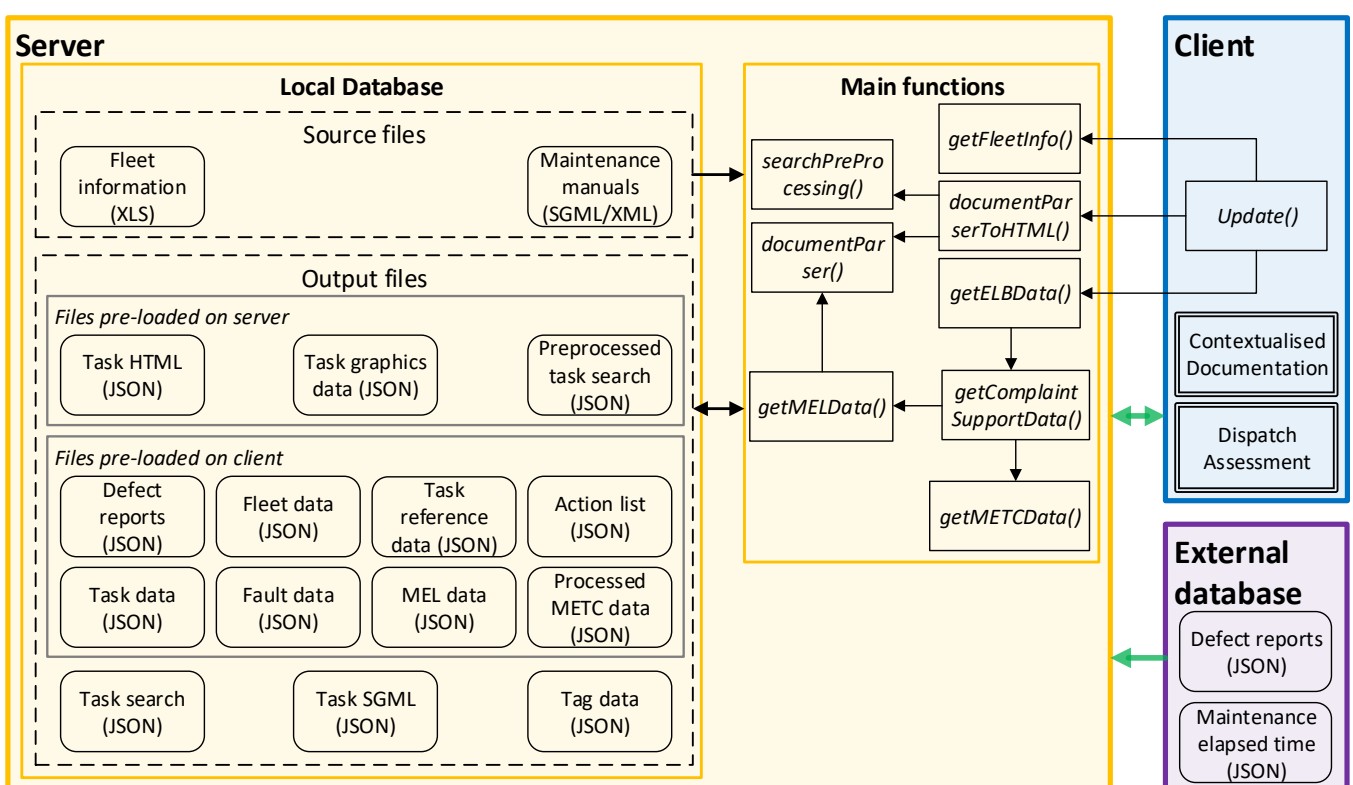

**Figure 1.** Prototype system architecture.

2.2.1. Server

The server, located in Figure 1 at the left side of the system architecture in yellow, is hosted on a Virtual Private Server (VPS) and has two main elements: a local database and the main functions. The local database refers to the internal storage on the VPS and consists of source files and output files. The sources files solely act as input files for the main functions, indicated by the unidirectional black arrow to the main functions, whereas the generated output files are also reused by other main functions, as indicated by the

bidirectional black arrow. Several output files are pre-loaded either on the server (3) or client (8) for efficient access during the use of the prototype, because these files are accessed frequently. The main functions as well as their interaction with different data sources and other functions are explained in detail in Section 2.3.

### 2.2.2. Client

The client is located at the right side of the system architecture in Figure 1 and communicates bidirectionally with the server, based on the socket.io package. The client features two integrated applications: contextualized documentation and dispatch assessment. Contextualized documentation provides access to relevant maintenance tasks during maintenance execution. Dispatch assessment provides the decision support during aircraft dispatch and is detailed in the case study in Section 3. Administrators also have access to the *Update()* function, which initiates a sequence of functions on the server to update the decision support and task support information to the latest information available.

The prototype user interface has been developed after careful analysis of the information required for dispatch decision-making, combined with the feedback from technicians during multiple design iterations. As a result, the final user interface limits the amount of information displayed to essential decision support information (see Figure 2), aiming to provide efficient decision support to technicians and minimize the cognitive effort for using the system.

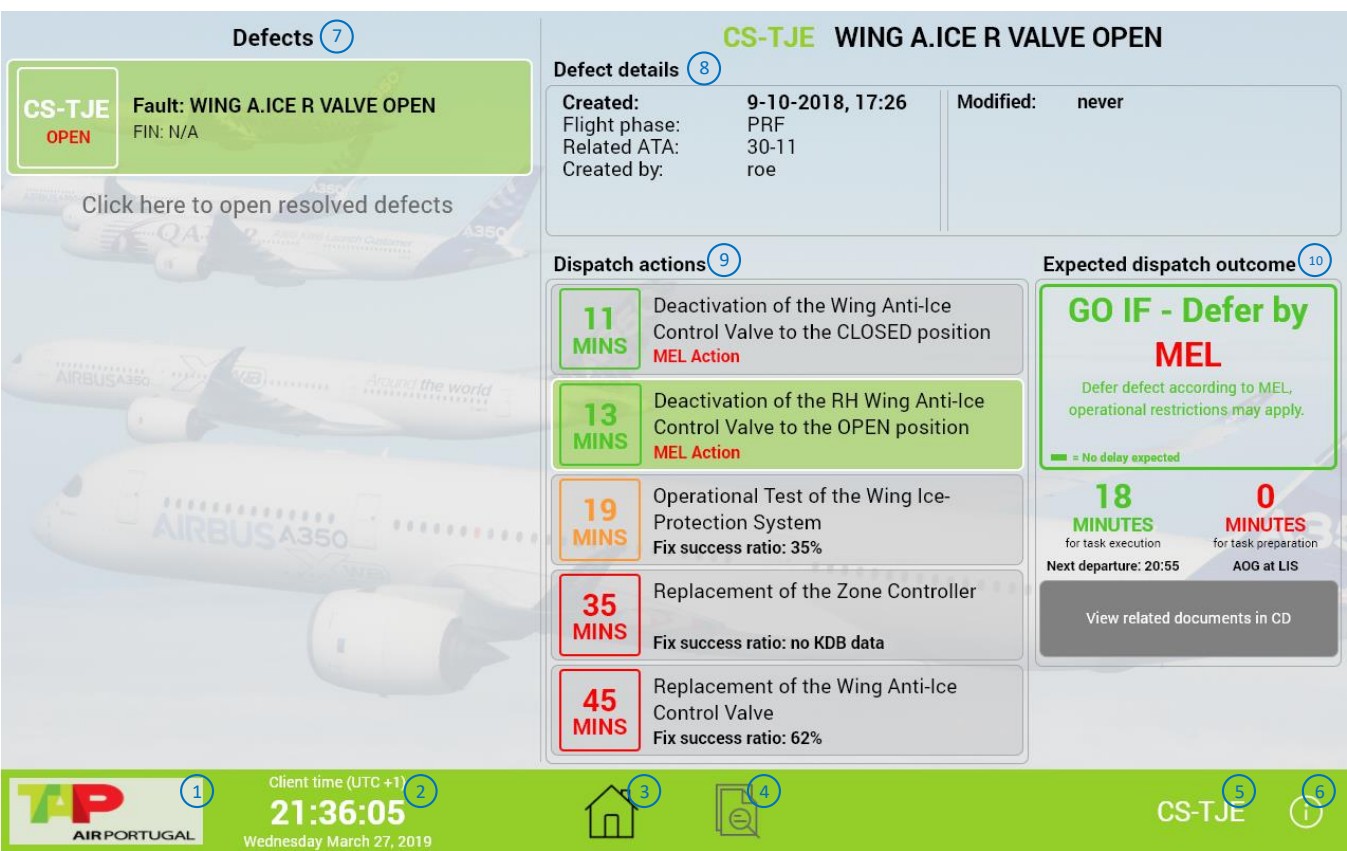

**Figure 2.** Example of the dispatch assessment prototype screen.

### 2.2.3. External Database

The external database, in purple at the lower right of the system architecture, consists of data generated by applications of AIRMES project partners: defect reports created in the Electronic Logbook (ELB) and maintenance elapsed time data from the Maintenance Elapsed Time Control (METC) application. This information is hosted on a cloud environment that has been developed to integrate AIRMES applications. The prototype can

retrieve this information by the *getELBData()* and *getMETCData()* functions through HTTP calls with the request package, available in the Node.js ecosystem, and is subsequently processed into new data objects.

*2.3. Key System Functions*

The key server side functions of the prototype automate the identification and retrieval of decision support information from the input data, and they output different data objects that provide fast access to relevant information for both client and server. Administrators can initiate the *Update()* function on the client, addressing system requirement **SR6**, which calls three functions that are executed server side: *getFleetInfo()*, *documentParserToHTML()*, and *getELBData()*. The *Update()* function is initiated manually in the prototype, but, technically, it can be automated when, for example, a new defect is reported or a new manual (revision) is uploaded.

2.3.1. getFleetInfo()

*Addresses system requirement*: **SR3**
*Input*: fleet information provided by the airline (XLS).
*Functionality*: this function transforms the fleet information provided by the airline, typically in XLS format, to an object to ensure efficient access. It contains information such as the tail number, the manufacturer, the model, and the Customer Serial Number (CSN) and is pre-loaded on the client at user login.
*Output*: fleet data (JavaScript Object Notation (JSON) is used to efficiently transmit structured data over a network connection).

2.3.2. documentParserToHTML()

*Addresses system requirements*: **SR1**, **SR2**
*Input*: maintenance manuals (SGML).
*Functionality*: this function retrieves relevant information from the maintenance manuals and stores this information in objects for efficient access. A requirement for this functionality is that maintenance manuals are available in their original electronic format (i.e., SGML), either on a local or a remote server. The maintenance manuals are processed one by one, and it starts with importing and subsequent pre-processing of the manual, i.e., removing redundant tags and replacing special characters. The next step sends the manual as a *string* to the document parser (i.e., *documentParser()*), which returns an array of objects containing detailed information about each tag. This array is then used to identify, retrieve, and store relevant information from the maintenance manual string, and, simultaneously, the original string is converted from SGML to HTML. This function generates three objects that are pre-loaded on the client at user login for fast access to the information:

- the task data object (JSON) contains metadata for each individual task in the manual, such as the task title, the effectiveness array (i.e., to which CSNs this task applies), the task ATA chapter and section, and the revision date. An example is shown in Figure 3. The task identification (i.e., based on the ATA numbering system) is used as the main object key;
- the task graphics object (JSON) contains metadata about graphics used in the manual, such as the number of sheets (i.e., a graphics reference can include multiple sheets), the associated illustrations with their reference and title, and the effectiveness array. The illustration files are located on the server and are retrieved when requested by the client. Illustrations are referenced by the graphics reference in the manual, which is therefore used as the object key in this task graphics object;
- the task HTML object (JSON) contains the HTML conversion of the original SGML-based maintenance task and is used to display the task content on the client (i.e., contextualized documentation). To maintain the original look and feel of the manuals, a cascading style sheet (CSS) is developed to control the layout. Similar to the task data object, the task HTML object also uses the task identification as the main key.

These objects are mainly used for the client-side functionality of dispatch assessment. To support the manual search function of contextualized documentation, a task search object is generated by this function as well. This object contains the string for each task, from which all the tags, double spaces, and tabs are removed. This object is sent to the *searchPreProcessing()* function for further pre-processing. Additionally, the SGML of each task is stored in an object array for later use in the *getComplaintSupportData()* function. After the initial execution of this update function, this information extraction from the manuals only needs to be performed when a new revision of a manual becomes available. When there are no new revisions of the manuals, the existing output, stored in the local database, is used.

*Output*: task data (JSON), task graphics data (JSON), task HTML (JSON), tasksearch object array (JSON), task SGML object array (JSON).

*Calls*: documentParser(), searchPreProcessing().

```
▼ AMM-30-11-00-040-001-A:
    ATACH: "30"
    ATASECT: "11"
  ▶ Effectiveness: ["ON ALL A/C"]
    Key: "EN30110004000100"
    ManualType: "AMM"
    Revision: "01-05-2018"
    Title: "Deactivation of the Wing Anti-Ice Control Valve to the CLOSED position"
  ▶ __proto__: Object
```

**Figure 3.** An example entry in the task data object, in this case for task "AMM-30-11-00-040-001-A".

### 2.3.3. documentParser()

*Addresses system requirements*: **SR1**, **SR2**

*Input*: maintenance manual (string, UTF-8).

*Functionality*: aircraft maintenance documents are SGML-based, which means that the information in these documents is semi-structured by using a standardized markup. A basic example of the SGML syntax is:

```
<title>Introduction to SGML</title>
```

Tags are enclosed by angle brackets and the closing tag is indicated by an additional (forward) slash. Tags contain contextual information about the content in between the tags (e.g., a title). Opening tags may also contain additional properties (or attributes) that provide additional contextual information. For example, by adding a "chapter" property, the document creators can indicate that this concerns the title of chapter "30":

```
<title chapter="30">
```

To be able to identify and extract information efficiently from the maintenance manuals, a custom-depth first parser was developed to process SGML-based documents, since this functionality was not readily available in the aircraft maintenance domain. After importing the manual as an UTF-8 encoded string, it is sent to the parser.

The processing steps performed by the parser are detailed in Algorithm 1. The parser processes the string from its start in a while loop (line 1) until it reaches the end of the input string. At line 2 and 3, the start and end index of the first tag in the remaining string (*currentString*) are determined. The *indexOf* method in JavaScript returns the position of the first occurrence of a specified value in a string and returns −1 if that value does not exist in the string. If there is still a tag to process, i.e., the start and end of a tag exist in *currentString* (line 4), the parser continues with evaluation of the tag content itself. After identifying the type of the current tag (i.e., self-closing tags (`<tag/>`) at line 5, closing tags (`</tag>`) at line 7, or opening tags (`<tag>`)), the *tagContent*, i.e., removing all angle brackets and forward slashes, is determined at lines 6, 9, and 10, respectively. At line 11, the parser checks if the current tag has additional properties by checking if the tag content contains a space at line 11. When true, the function *getTagProperties* is called at line 13 to

extract all properties from the *tagContent* string, returning an array of objects, and the tag name is determined at line 12. For tags without any additional properties, the tag name is determined at line 14. Once all the tag details are determined, they are stored in a *tagObject*, which is then pushed into the *outputArray* (lines 15 and 16) unless the current tag is a closing tag. The following information is stored in the *tagObject*, of which an example is shown in Figure 4:

- *startWithTag*: the index of the opening angle bracket (<) of the opening tag.
- *startWithOutTag*: the index of the closing angle bracket (>) of the opening tag. This value is increased by 1 to set the index *after* the closing angle bracket.
- *endWithTag*: the index of the opening angle bracket (<) of the closing tag. Initially, a value of −1 is assigned (i.e., non-existing), until it is updated when a corresponding closing tag is found.
- *endWithOutTag*: the index of the closing angle bracket (>) of the closing tag. This value is increased by 1 to set the index *after* the closing angle bracket. Initially, a value of −1 is assigned (i.e., non-existing), until it is updated when a corresponding closing tag is found.
- *tag*: this is the *tagName*—in this case, "EIN"—indicating that this part of the string contains a Functional Item Number (FIN). For example, FIN 10DL refers to the right-hand wing anti-ice valve.
- *tagProperties*: an array of objects with all the tag properties. In this example, there is one property, indicating that the type of this "EIN" is exact.
- *listlevel*: a value to determine the indentation of content in the HTML representation.

```json
{
  "startWithTag": 9704,
  "startWithOutTag": 9722,
  "endWithOutTag": 9726,
  "endWithTag": 9732,
  "tag": "EIN",
  "tagProperties": [
    {
      "propertyName": "TYPE",
      "propertyValue": [
        "EXACT"
      ]
    }
  ],
  "listlevel": 19
},
```

**Figure 4.** An example of a *tagObject* entry in the *outputArray*.

---

**Algorithm 1:** Depth-first parser algorithm for SGML-based documentation.

---

**input** : A string, UTF-8 (*inputString*)
**output**: tag data object array, JSON (*outputArray*)

*outputArray* ⟵ []
*currentString* ⟵ *inputString*
*currentStringOffset* ⟵ 0

**1  while** *currentString.length* > 0 **do**

**2**　　*startIndexCurrentTag* ⟵ *currentString.indexOf*(*'<'*);

**3**　　*endIndexCurrentTag* ⟵ *currentString.indexOf*(*'>'*);

　　　/* determine if there is still a tag to process                                                */

**4**　　**if** *startIndexCurrentTag* > − 1 *&&* *endIndexCurrentTag* > − 1 **then**

　　　　　/* determine the tag type and the *tagContent*                                       */

**5**　　　**if** *the tag is self-closing (i.e., tag contains '/>')* **then**

**6**　　　　*tagContent* = *currentString.substring*((*startIndexCurrentTag* + 1), (*endIndexCurrentTag* − 1));

**7**　　　**else if** *the tag is a closing tag (i.e., tag contains '</')* **then**

**8**　　　　*closingTag* ⟵ *true*;

**9**　　　　*tagContent* = *currentString.substring*((*startIndexCurrentTag* + 2), (*endIndexCurrentTag*));

　　　　**else**

**10**　　　　*tagContent* = *currentString.substring*((*startIndexCurrentTag* + 1), (*endIndexCurrentTag*));

　　　　**end**

　　　/* extract additional tag properties (i.e., when *tagContent* contains a space) and
　　　　determine the tag name (e.g., TASK, TITLE, EFFECT)                                 */

**11**　　　**if** *tagContent.indexOf*(*' '*) > −1 **then**

**12**　　　　*tagName* = *tagContent.substring*(0, *tagContent.indexOf*(*' '*);

　　　　　/* call a function to get the tag properties from the *tagContent* string       */

**13**　　　　*tagProperties* ⟵ *getTagProperties*(*tagContent*) (returns an array of objects);

　　　　**else**

**14**　　　　*tagName* ⟵ *tagContent*;

　　　　**end**

　　　/* store the tag information in *tagObject* (i.e., contains indices, tag name and tag
　　　　properties) and push it into *outputArray*                                           */

**15**　　　**if** !*closingTag* **then**

**16**　　　　*outputArray.push*(*tagObject*);

　　　　**else**

　　　　　/* traverse backward in the existing *outputArray* to check for a corresponding
　　　　　　opening tag (i.e., with the same *tagName*)                                    */

**17**　　　　**for (** *var arrayEntry* = (*outputArray.length* − 1); *arrayEntry* ≥ 0; *arrayEntry* − − **) {**

**18**　　　　　**if** *tagName* == *outputArray*[*arrayEntry*][*tagName*] **then**

　　　　　　　/* replace the start and end indices of the closing tag                 */

**19**　　　　　　*outputArray*[*arrayEntry*][*tagName*][*endWithOutTag*] ⟵ *startIndexCurrentTag*;

**20**　　　　　　*outputArray*[*arrayEntry*][*tagName*][*endWithTag*] ⟵ *endIndexCurrentTag* + 1;

　　　　　　　/* break off the loop                                                          */

**21**　　　　　　break;

　　　　　**end**

　　　　**}**

　　　　**end**

　　　/* remove all information before the *next* tag from *currentString*              */

**22**　　　*currentString* = *currentString.substring*(*startIndexNextTag*);

　　　/* update *currentStringOffset*                                                        */

**23**　　　*currentStringOffset* += *startIndexNextTag*;

　　**end**

　**end**

**24** return *outputArray*;

---

For closing tags, the algorithm traverses backwards in the existing *outputArray* (line 17) to find the first *tagObject* with a similar *tagName* (line 18) from the end of the array. This is necessary to find the correct corresponding opening tag in the case of multiple consecutive opening tags with the same *tagName*. An example of this situation is as follows:

`<title><title>`Content`</title></title>`

Here, matching colors indicate which opening tags correspond to which closing tags. If the algorithm would traverse forward through the *outputArray*, it would consider the red closing tag to be related to the blue opening tag, since that would be the first corresponding *tagName* in the *outputArray*. This is avoided by traversing through the *outputArray* backward, ensuring that the closing tag is always related to the last opening tag with the same *tagName*. If the values for *endWithOutTag* and *endWithTag* in the originally stored opening *tagObject* are still −1, these values are overwritten by the start (line 19) and end (line 20) indices of the current closing tag (*startIndexCurrentTag* and *endIndexCurrentTag* + 1, respectively). This additional check avoids overwriting of the red *tagObject* information when the blue closing tag is evaluated. If the corresponding opening tag information is updated, the for loop is ended at line 21. At line 22, the algorithm removes the currently evaluated tag and any content before the next tag in the string, thereby updating *currentString*. Line 23 updates the offset of *currentString* with respect to the original input string, which is relevant when the tag indices are stored in the *tagObject*. When the entire input string is processed, the algorithm returns the *outputArray* to the function that it was called by (e.g., *documentParserToHTML()*).
*Output*: tag data object array (JSON).

### 2.3.4. searchPreProcessing()

*Addresses system requirement*: **SR3**
*Input*: task search object array (JSON).
*Functionality*: the function *documentParserToHTML()* creates an array of objects, one object for each task in the manual. An object contains metadata for each task (e.g., task number, task title) as well as the entire task string from which all tags are removed. This function prepares the content string for the search functionality of contextualized documentation by noise removal (e.g., removal of special characters, excessive spaces, internal references to warnings and cautions, and trimming) and normalization (i.e., converting all characters to lowercase). The results are stored in an array of objects similar to the input, one object per maintenance task, but with the contents pre-processed for searching.
*Output*: pre-processed search object array (JSON).

### 2.3.5. getELBData()

*Addresses system requirement*: **SR4**
*Input*: defect reports (JSON).
*Functionality*: this function retrieves defect reports that are available on an external server. The defect reports are created in the ELB and contain information such as the affected tail number, a unique defect identifier, creation date, status, and the fault message. Each defect report is stored as an object in an array.
*Output*: defect report object array (JSON).
*Calls*: *getComplaintSupportData()*.

### 2.3.6. getComplaintSupportData()

*Addresses system requirements*: **SR4**, **SR5**
*Input*: defect report object array (JSON), selected part of the Troubleshooting Manual (TSM) (SGML), TSM per task (SGML)
*Functionality*: this function retrieves decision support information from various resources, based on unique fault messages in the defect report object array. The process has four

distinct steps: (1) identify which troubleshooting tasks are related, (2) identify the available maintenance alternatives per troubleshooting task, (3) identify deferral alternatives, and (4) generate a list of all dispatch options and gather decision criteria information.

(1) **Identify related troubleshooting tasks**

The TSM contains a section that lists fault messages and their related troubleshooting task(s). In order to automatically retrieve this information, this TSM section is sent to the *documentParser()* and, by combining the *outputArray* with the SGML of the TSM section, the relevant TSM tasks for each unique fault message are determined. Moreover, the process collects additional metadata for the given fault message from this TSM section, such as the ATA chapter and section numbers, the revision date of the TSM task, and its effectiveness (the effectiveness or applicability of a (part of a) maintenance task is determined by the Customer Serial Number (CSN), a unique identifier for an aircraft).

(2) **Identify maintenance alternatives per troubleshooting task**

Every troubleshooting task contains a table with a list of maintenance alternatives (i.e., tasks in other maintenance manuals) that potentially resolve the problem. Again, the *documentParser()* is called and now combined with the SGML per task (output of *documentParserToHTML()*) for each identified troubleshooting task, resulting in an object for each troubleshooting task with references to maintenance tasks, including their titles and effectiveness.

(3) **Identify deferral alternatives**

Instead of permanently resolving a problem, some faults allow for a so-called "dispatch by MEL". The Minimum Equipment List (MEL) details which parts or systems of an aircraft may be inoperative, for how long, and if such deactivation leads to specific operational limitations. Deferring a problem is applied regularly in current daily operations and allows shifting of a rectifying maintenance action (such as a permanent repair) to a more convenient time and location (i.e., not during flight operations and at the maintenance facility), without compromising safety. The available deferral alternatives, if any, are identified by the *getMELData()* function detailed in Section 2.3.7. The returned information includes any references to other maintenance manuals, such as the Aircraft Maintenance Manual (AMM), because, typically, a deferral alternative also requires some type of maintenance execution—for example, the deactivation of a part.

(4) **Merge alternatives and retrieve decision criteria information**

Here, all identified (unique) maintenance and deferral alternatives are combined into one dispatch options list and the decision criteria information associated with individual alternatives can be retrieved. Within the scope of this prototype, the decision criterion is time—in particular, the time available before the next departure and the time required to complete a maintenance task. The mobile METC application records the task execution time by having the technician confirm the start and end time of task execution on their mobile phone. While some maintenance manuals do provide this type of information, practitioners indicate that these indications are rarely accurate. The METC application provides realistic information about task execution times that can be used for optimized planning and progress monitoring. The METC data are stored on an external server and retrieved for every dispatch option by the *getMETCData()* function detailed in Section 2.3.8, having the maintenance task identifier as input. Once the METC information is returned, a final step in this process is to combine any removal and installation tasks. These are technically seen as separate alternatives but, in practice, are linked and executed in sequence. These combinations are identified and combined, including references to individual maintenance tasks and the METC information.

Once all processes are completed, the *getComplaintSupportData()* function has generated three objects: an object that contains information related to each unique fault message in the defect reports (e.g., the associated TSM tasks, ATA reference), an object with the

maintenance alternatives identified in each TSM task, and an object with the combined list of dispatch options per fault message, including their estimated execution times.
*Output*: fault data (JSON), task reference data (JSON), action list (JSON).
*Calls*: *documentParser()*, *getMELData()*, *getMETCData()*.

### 2.3.7. getMELData()

*Addresses system requirement*: **SR4**
*Input*: Minimum Equipment List (MEL) (XML), fault data (JSON).
*Functionality*: this function processes the MEL and retrieves relevant information, which is stored in an object and subsequently sent back to the function that it was called by. The first step in this function is loading the MEL (available as XML) and sending it to the *documentParser()*. The returned *outputArray* is then combined with the XML string to extract information based on the ATA chapter and section numbers that are related to the fault message. Amongst the information that is retrieved and stored for each fault message is: the MEL task identifier, the title of the MEL task, the related maintenance task identifier, the repair interval, and the applicability based on the Manufacturer Serial Number (MSN).
*Output*: MEL data (JSON).
*Calls*: *documentParser()*.

### 2.3.8. getMETCData()

*Addresses system requirement*: **SR4**
*Input*: maintenance task identifier (string, UTF-8), maintenance elapsed time data (JSON).
*Functionality*: to assess the feasibility of a given dispatch option with respect to the scheduled departure time, METC data are used to determine the average task duration. For each maintenance task identifier, the external METC database is checked for data entries (i.e., logged timestamps for maintenance tasks), and, subsequently, the task duration is calculated. In this process, task durations that clearly result from faulty measurements are excluded—for example, when a technician forgets to close the task. The range of valid measurements for a given (range of) task(s) is determined in consultation with industry experts. Using only valid measurements, the average task duration in seconds is calculated and stored in an object together with the maintenance task identifier string.
*Output*: METC processed data (JSON).

## 3. Case Study

Having implemented the main functions and the client user interface using the Node.js platform, all system requirements are addressed. This section discusses the validation of the prototype for dispatch decision support with respect to the functional requirements introduced in Section 2.1.

### 3.1. Demonstration Setup and Scope

To address the issues that technicians face during dispatch assessment and reduce the resulting technically induced flight disruptions, a case study has been performed to evaluate the prototype. To have a realistic representation of an operational scenario that is suitable for prototype evaluation, industry experts selected a use case that is not too easy (i.e., technicians could rely on their experience to resolve the issue and not use the prototype), but not too complicated either (i.e., technicians would focus more on the maintenance task itself than on the use of the prototype). The fault message "WING ANTI ICE R VALVE OPEN" on an Airbus A321-211 was determined to be a suitable use case. This is a warning message that is displayed on the on-board electronic centralized aircraft monitor (ECAM) and indicates that there is an issue with the right-hand wing anti-ice valve (WAIV), which is part of the ice protection system that prevents the accumulation of ice on the leading edge of the wing. Accumulation of ice on the wing heavily deteriorates the handling performance and the wing's ability to generate lift, which can lead to catastrophic events (e.g., [34,35]). All project partners adopted this use case, for

which a 3-day demonstration storyboard was created, of which the outline is shown in Figure 5.

Experienced technicians then executed the demonstration storyboard in an operational maintenance facility in Lisbon. The prototype was used by multiple technicians, both on-site through the Android app and in the office using a web browser on a desktop computer. It is noteworthy that, in reality, the three days referred to in the storyboard do not need to be consecutive calendar days—an item may be deferred for multiple days—and that the demonstration has been performed on a single day to showcase the seamless integration of the different developed technologies within the AIRMES project. To limit the processing time, only ATA chapter 30, containing the maintenance tasks related to this particular fault, of the AMM and TSM were included. The MEL is less extensive and was included in its entirety.

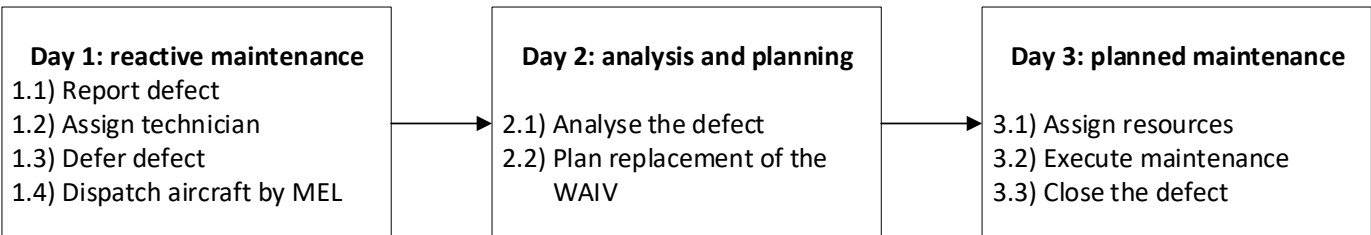

**Figure 5.** Storyboard outline for the final prototype demonstration.

### 3.2. Demonstration Execution

The demonstration starts at step 1.1, when the pilot reports a defect in the aircraft-attached ELB, which is then uploaded to a ground server. At step 1.2, a technician is assigned to the aircraft and will drive from the office to the apron. Meanwhile, the defect report is retrieved by the *getELBData()* function, after which the subsequent functions to retrieve task-specific information (e.g., *getComplaintSupportData()*) are executed. These functions require less than a minute to complete and all information to perform the dispatch assessment will be available by the time the technician arrives at the aircraft. Figure 2 is an example of the information that both the technician at the aircraft as well as other remote users (e.g., Maintenance Control Center (MCC), Operations Control Center (OCC)) have access to, through the mobile app or web browser, respectively.

### 3.2.1. Prototype User Interface

In Figure 2, the key items are indicated by dark blue, numbered circles. The main menu items are located at the bottom, from left to right:

1. The currently logged-in user. The color scheme of the layout is adjusted to the user.
2. The current local time and date, including an indication of the local time zone offset with respect to the Coordinated Universal Time (UTC).
3. The home menu button that directly provides access to the dispatch assessment functionality (currently selected).
4. The direct access button to contextualized documentation.
5. The tail number associated with the currently selected defect (in item 7). Clicking this button provides access to aircraft details (e.g., model, CSN) and the flight schedule.
6. An information button providing access to background information, such as contact information, the current prototype version, and the project website.

On the left side of the screen, at item 7, all current unresolved defects are displayed. The currently only unresolved defect is selected, which results in displaying the information on the right side of the screen. At item 8, details of the reported defect are displayed, such as the creation date and related ATA section. The right side of this section displays information about previous deferral or modification attempts, which, in this case, displays "never", this being a newly created defect report. At item 9, the dispatch actions

(i.e., alternatives) are displayed, which follows from the action list created by the *getComplaintSupportData()* function.

### 3.2.2. Comparing Dispatch Options

For the current fault, there are six possible alternatives that are retrieved from the AMM and MEL, but only five are shown due to the fact that one alternative is filtered out because of the effectiveness check (i.e., the alternative does not apply to this particular CSN). The alternatives are ranked such that deferral alternatives are at the top of the list (by operator preference, indicated by "MEL Action") and then by the expected average time to complete (displayed in the colored, square boxes), as calculated by the *getMETCData()* function. After consultation with industry experts, the lower and upper boundaries for valid METC measurements for this case study are set to 3 min and 8 h, respectively. The alternatives are color coded green, yellow, or red, based on the time to complete them with respect to the time available before the next departure. The system automatically updates the color coding; during the demonstration, an update frequency of 30 s was used. For example, the option "Operational Test of the Wing Ice-Protection System" in Figure 6 just turned yellow because there are only 18 min before the next departure, while the task requires 19 min. Additionally, for alternatives that can permanently resolve the defect, thus excluding the deferral options, the metric "Fix success ratio" is shown. This metric indicates how successful that alternative historically has been with respect to resolving the current defect for similar aircraft models, and could aid the technician in decision-making. However, due to a lack of data in the knowledge database—both in quantity and quality—this metric has not been included in the final demonstration. Once an alternative is selected, the expected dispatch outcome information is shown (item 10). Here, the following information is shown: the dispatch type (i.e., in this case a MEL deferral), the time available before the next departure, any additional time to prepare a task when a fault is already reported during flight (i.e., indicating 0 min, because the aircraft has already landed), and a link to contextualized documentation that will only show the maintenance tasks related to this alternative, as shown in Figure 6. It shows four distinct items, by indicated dark blue, numbered circles:

1. the information container showing the current defect and selected alternative;
2. the contextualized documents (3 in total, with the related AMM task selected);
3. metadata of the selected document that will remain visible on top when scrolling;
4. the actual content of the AMM maintenance task. When no alternative is selected, the link to contextualized documentation will show all maintenance tasks related to all alternatives.

### 3.2.3. Dispatch Decision-Making

Technicians can easily compare different dispatch options and, combined with their expertise, select the best alternative for the given situation. In agreement with the pilot in command—who is ultimately responsible for the safety during flight and always has a final say in this decision—the technician decides to defer the defect and dispatch the aircraft by MEL, after performing the required maintenance task, following step 1.3 and 1.4 of the demonstration storyboard. Continuing to day 2 of the storyboard, the MCC analyzes the defect in the back office to determine the best dispatch option leading to a permanent fix and subsequently plan the maintenance, while the aircraft can safely continue flight operations. Amongst the information accessed during this process are the related maintenance documents available through contextualized documentation. At day 3, a technician is assigned to replace the WAIV, having the task-specific information available through the prototype during maintenance execution. The demonstration is concluded by closing the defect in the electronic logbook.

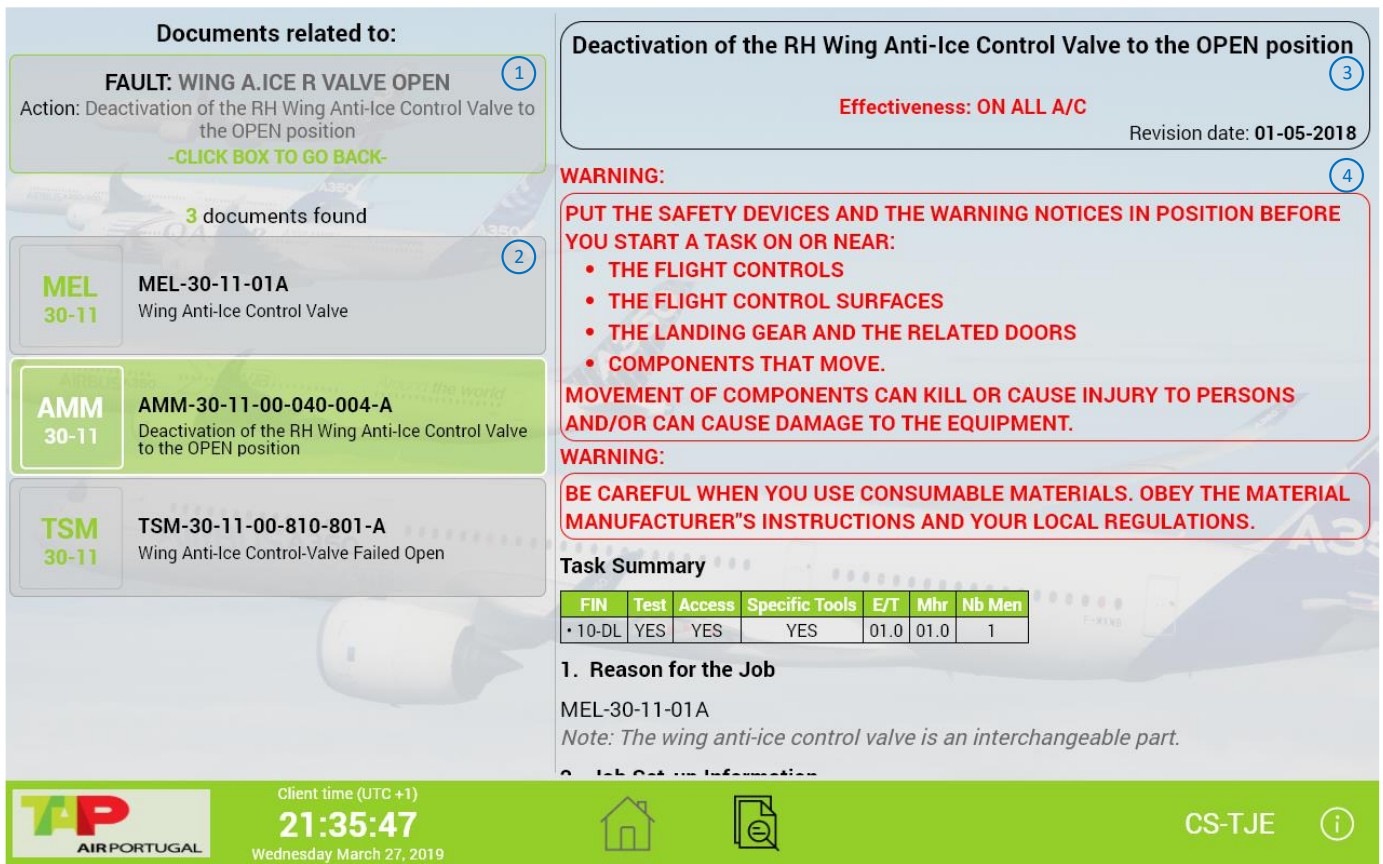

**Figure 6.** Example of the contextualized documents with respect to the alternative.

## 4. Results and Discussion

Although technicians have to switch between multiple applications during the demonstration, the integration between applications ensures a seamless user workflow. With respect to the impact of the dispatch assessment prototype, and the aim to reduce technically induced flight reductions, the demonstration shows that the prototype enables technicians to make well-informed decisions while reducing the decision time significantly. Furthermore, the demonstration shows that the process of identification and retrieval of the dispatch options and its related documents from different maintenance manuals is performed correctly, i.e., showing the right and complete information with respect to the fault message, while taking into account the effectiveness of individual tasks. Verification of this process was done by a manual lookup of the dispatch options and related task-support data, and comparing the result with the automatically generated results by the prototype. When solely looking at the time required to make a dispatch decision, technicians indicate that this can be reduced from 5–10 min in the current situation to just 15 s by using the prototype. Moreover, technicians indicate that they currently have a predominant tendency to fix the aircraft (i.e., only consider final fix alternatives) and that they more promptly consider deferral options by using the prototype, due to the concise overview of all available alternatives. As such, the prototype allows technicians to make better-informed decisions. Another major time-saving aspect is that with all the information available at the aircraft, technicians no longer need to drive back and forth to acquire information, saving valuable time in the turnaround. This can lead to more feasible dispatch options that will not result in a delay, because the time previously spent on logistics can now be used to perform actual maintenance. Technicians estimate that the combined time savings of the automated alternative identification and having the information available at the aircraft can reduce the time that they spend on the dispatch process from 15 min to 15 s on average (i.e., 98% reduction), thus significantly reducing the chance of technically induced delays. The fact

that the prototype reduces the workload is an important factor for technicians in adopting the solution, as they stated that previously introduced "aids" only increased the workload without providing any clear benefit, and thus were left unused. Finally, converting to a digital maintenance information process also reduces the waste of paper (i.e., the current process is mostly paper-based) and the number of movements (i.e., technicians driving back and forth to acquire information), thereby reducing the environmental footprint of the process.

*Achievement of the Functional Requirements*

In Section 2.1, we defined several functional requirements that would be achieved if the system requirements **SR1** to **SR8** are successfully implemented. Using the demonstration results for verification and validation, the following conclusions can be drawn with respect to the functional requirements:

**FR1** *Provide technicians with an concise overview of the available dispatch options, following from a fault message, including their expected outcome with respect to the flight schedule.*
The prototype fully meets this requirement by providing an intuitive overview of all possible and applicable maintenance actions related to a fault message, and, for each action, it displays the expected operational impact.
**FR2** *Provide technicians with access to task-support information.*
By providing direct access to related maintenance tasks through contextualized documentation, technicians can easily consult task-support information. Hence, this requirement is achieved.
**FR3** *Provide technicians with the most recent information to ensure safety compliance and accuracy.*
This functional requirement is considered partly met, due to the fact that the current update process needs to be initiated manually. It is technically feasible to automate the update process whenever any of the source data are changed—for example, when a new defect is reported or when a new manual revision is uploaded.
**FR4** *Provide technicians with fast (i.e., <1 min) access to relevant maintenance information anywhere.*
This requirement is achieved. Technicians have near-instant access to dispatch decision support information and task-support information wherever a (wireless) internet connection is available.
**FR5** *Provide external applications access to maintenance documentation.*
Fully achieved by providing access to maintenance documents on task level through a standard link in combination with the task identifier. Moreover, external applications can directly access dispatch assessment with a specific defect selected, which was required for a seamless workflow.

## 5. Conclusions

This paper presents a novel web-based prototype decision support system for aircraft dispatch, addressing two gaps identified in the literature:

1.  a lack of published research for operational decision-making in the (aircraft) maintenance domain, by introducing a novel system architecture and its subsequent implementation by means of a prototype to provide real-time decision support;
2.  the indicated lack of correlation between theory and practice in decision support systems research, by evaluation of the prototype through a case study in an operational environment and showing its practical relevance.

The prototype addresses multiple issues that technicians currently face during dispatch assessment, mostly related to weaknesses of maintenance documentation. This has led to several gains in aircraft (line) maintenance operations:

*   The case study indicates that the decision time for aircraft dispatch can be reduced up to 98% by fully automating the information retrieval, alternative identification and evaluation, and subsequently providing the technician at the apron with relevant decision support information.

- The web-based implementation enables integration of multiple data sources and provides access to maintenance information for informed decision-making with almost no geographical limitations.
- Next to reducing technically induced delays, an additional benefit is the reduction of environmental impact by eliminating the use of paper and fuel consumption in current maintenance operations.

During the demonstration, technicians were very positive about the ease of use of the prototype and they highly valued its practical use. As a result, our industry partner is currently considering further development of the prototype to a production-grade application.

*Future Research Directions*

While the prototype demonstration was very successful, it has limitations that can be addressed in future research. A forthright approach would be expanding the scope from a single failure to more or all potential issues by including a broader selection of maintenance manual content. It should be noted that updating the manuals is relatively time-consuming, depending on the processing capabilities of the server. This can be further optimized by iterative updates, i.e., only update parts of the manuals that have changed by checking the revision dates of sections or tasks. By expanding the scope, the manual verification of completeness and correctness of the presented dispatch options and related task-support information per fault message is no longer feasible. Hence, approaches for automated verification should be investigated.

To enable the use of the system without an electronic logbook, the entire range of fault messages in the troubleshooting manual can be processed independently of any reported defect. Thus, one can run step 1 of the *getComplaintSupportData()* function without the defect report object array as input and output (i.e., fault data object, task reference data object, action list object) for all fault messages in the troubleshooting manual. This allows for manual selection of any known fault in the prototype interface, instead of only the (electronically) reported defects.

Another research direction is to include more decision criteria besides time and implement multi-criteria decision-making models. Some examples of such criteria are: the current location of the aircraft (i.e., maintenance is preferably performed at the home base), the availability of spare parts to determine the practical feasibility of alternatives, and the inclusion of recovery models to include the cost (e.g., Hu et al. [36]) and consequences (e.g., propagated network delay as modelled by Santos et al. [14] and Pyrgiotis et al. [37]) of choosing an alternative that would cause delay.

**Author Contributions:** Conceptualization, H.K., W.J.C.V. and R.C.; methodology, H.K. and W.J.C.V.; software, H.K.; validation, H.K. and W.J.C.V.; investigation, H.K. and W.J.C.V.; data curation, H.K.; writing—original draft preparation, H.K.; writing—review and editing, W.J.C.V. and R.C.; visualization, H.K.; supervision, W.J.C.V. and R.C.; project administration, H.K. and W.J.C.V.; funding acquisition, W.J.C.V. and R.C. All authors have read and agreed to the published version of the manuscript.

**Funding:** This project has received funding from the Clean Sky 2 Joint Undertaking under the European Union's Horizon 2020 research and innovation programme under grant agreement no. 681858, AIRMES project. For more information, see the project websites: http://www.airmes-project.eu (accessed on 11 September 2019) and https://www.cleansky.eu/ (accessed on 11 September 2019).

**Institutional Review Board Statement:** Not applicable.

**Informed Consent Statement:** Not applicable.

**Data Availability Statement:** No new data were created or analyzed in this study. Data sharing is not applicable to this article.

**Acknowledgments:** We would like to thank TAP Air Portugal for sharing their expertise, providing us with maintenance data, and granting us access to their technicians and maintenance facilities in Lisbon for prototype demonstration and evaluation.

**Conflicts of Interest:** The authors declare no conflict of interest. The funders had no role in the design of the study; in the collection, analyses, or interpretation of data; in the writing of the manuscript, or in the decision to publish the results.

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
