# Peer review of "A Web-Based Decision Support System for Aircraft Dispatch and Maintenance"

_aerospace, doi:10.3390/aerospace8060154_

Round 1

Reviewer 1 Report

This paper introduces a novel web-based prototype decision support system to aid technicians during aircraft dispatch decision making and subsequent maintenance execution. A system architecture for real time dispatch decision support is established and implemented. The developed system is evaluated through a case study in an operational environment by licensed maintenance technicians. The entire paper is well written and structed. However, there are some issues need to be addressed before it can be published:

  1. In the paper, the authors claim the proposed decision support system can be used for aircraft dispatch and maintenance. Does it support the IMR of aeroengine? If not, the reviewer recommends making a few comments on this aspect in the paper. The following articles could be used as references:
    • Alatorre, D., Nasser, B., Rabani, A., Nagy-Sochacki, A., Dong, X., Axinte, D. and Kell, J., 2018. Teleoperated, in situ repair of an aeroengine: Overcoming the internet latency hurdle. IEEE Robotics & Automation Magazine26(1), pp.10-20.
    • Alatorre, D., Nasser, B., Rabani, A., Nagy-Sochacki, A., Dong, X., Axinte, D. and Kell, J., 2018, October. Robotic boreblending: the future of in-situ gas turbine repair. In 2018 IEEE/RSJ International Conference on Intelligent Robots and Systems (IROS)(pp. 1401-1406). IEEE.
    • Wang, M., Dong, X., Ba, W., Mohammad, A., Axinte, D. and Norton, A., 2021. Design, modelling and validation of a novel extra slender continuum robot for in-situ inspection and repair in aeroengine. Robotics and Computer-Integrated Manufacturing67, p.102054.
  2. The reviewer fully appreciate the engineering contribution of the presented research. But, what is the scientific contribution of this work to the communities? Can the authors make some comments at the beginning of chapter II.

Overall, this paper is interesting and well written. However, the identified issues need to be addressed before it can be published.

Author Response

Dear reviewer,

Thank you for your time and effort in reviewing our submission. We appreciate your constructive feedback and have addressed your concerns as follows:

  • In the paper, the authors claim the proposed decision support system can be used for aircraft dispatch and maintenance. Does it support the IMR of aeroengine?

The system does support the inspection, maintenance and repair of engines. The maintenance support information it provides is an exact copy of common manuals such as the Aircraft Maintenance Manual and Troubleshooting Manuals, which include the ATA chapters (70 and 80) on power plants. However, for this research we limited the scope to the ice and rain protection (ATA 30).

  • The reviewer fully appreciate the engineering contribution of the presented research. But, what is the scientific contribution of this work to the communities? Can the authors make some comments at the beginning of chapter II.

The contribution, both scientific and practical, are stated in $1.3. To further clarify the theoretical contribution we have added the following to $1.3: “The approach facilitates straightforward combination of multiple heterogeneous information sources and is the first to address operational decision making for unexpected failures in the aircraft maintenance domain.”

We have indicated changes by red text.

Reviewer 2 Report

Thanks for this very interesting paper. Attached are my suggestions.

Author Response

Dear reviewer,

Thank you for your time and effort for this elaborate review. We highly appreciate your constructive and detailed feedback and believe addressing these points will increase the quality of our submission. We addressed your points as follows:

  • Lines 29-30 : it seems that you assimilate CBM and data-driven PM. PM is indeed within the scope of CBM, but not the only one. Maintenance on early warning, without prediction or data analysis, is for example also part of CBM. You can adjust the text to avoid this assimilation.

This was indeed an unintended assimilation of CBM and PM, as CBM indeed does not need to be data-driven. The point we tried to make was a shift towards data-driven approaches. We adjusted the two sentences to clarify the statement.

  • Line 105 (§2.1) + line 206 (§2.3.2): Is there an assumption that all Maintenance Manuals to be imported in your app shall be an electronic format ? Structured according to a specific format such as S1000D? Could clarify those points? I mean that an airline that wants to fully use you app shall fully switch to paperless doc (e.g. no manual tech logbook). + What about airlines that work with manufacturer online tech pub? The app is compatible with that way of working?

Yes, the assumption is that the maintenance manuals are available in their original electronic format (SGML). These documents are structured according to the ATA iSpec 2200 or S1000D standards. We have added a sentence in $2.3.2 to clarify this. In the section for future research ($5.2, line ~635) we suggest to extract all fault messages from the troubleshooting manual. That way, a technician can manually select a fault message in the prototype DSS to get decision support, without the need to have an electronic logbook (ELB). Because indeed an ELB is currently not that common in aircraft maintenance operations. The app is perfectly able to connect to other applications such as tech pubs. Here, the problems is probably data-ownership, as manufacturers are not keen on sharing source files. In our situation we could acquire the original electronic manuals as SGML, because the partner airline had ownership over these documents.

  • Lines 198 (§2.3.1) + 226 + 550: effectiveness / applicability of maintenance tasks and data are very related to aircraft applied configuration, that is not only a/c tail number. In your text, it seems that effectiveness information relies only on tail number. The reality being more complex that that (e.g. modifications, equipement P/N), can you clarify the way you evaluate effectiveness of task? Important in my mind as in lines 499-500, you mention that some task are filtered out of the info presented to maintenance technician according to this evaluation. My concern : how to ensure that the info presented to the technician is exhaustive and correct, and that the filtering not leads to incorrect decision.

Within the scope of this research we indeed applied the effectiveness based on the Customer Serial Number. This serial number is specific to the initial customer – and different from the Manufacturer Serial Number- and holds into account the original configuration of the aircraft (incl. part manufacturers). Within the manuals, which are frequently updated and adjusted to the customer, the manufacturer indicates by CSN which sections (on task level) are relevant to a specific aircraft. As long as the maintenance history and modifications are kept up to date, and the manuals adequately updated, the information presented should be exhaustive and correct. Moreover, when the technician consults a related maintenance task, the applicability and revision date are always clearly displayed (see Figure 6). Technicians indicated this as essential information. Alternatively, technicians can always browse all (related and unrelated) maintenance tasks through another function of the app (item 4 in Figure 5). Future research regarding this point could also consider the integration of information from an aircraft configuration management system.

  • Line 369 and following: even though it is referred to during line maintenance, MEL is an OPS document, not a maintenance part. It relates to pilot-in-command decision, and not maintenance decision. I fully understand why you mention MEL, but for precise wording and to be accordance with regulatory processes, I recommend that you adjust the text to reflect that point.

Agreed. Although not specifically stated in this paper, to our knowledge, the pilot-in-command can always reject a proposed maintenance alternative. Because ultimately (s)he is responsible for the safety of the aircraft and its passengers and crew. Hence, a technician will always consult the pilot-in-command with respect to the proposed solution. We changed the sentence on line 533 to clarify the role of the pilot-in-command and where appropriate we have replaced “maintenance alternative” with “dispatch option” throughout the document.

  • Line 369 and following: do you also consider CDL? Or does your app focuses on system failures with failure code? Can you clarify?

No, within the scope of this research we did not include the configuration deviation list. The scope for the case study was limited to fault messages from the ECAM regarding ice and rain protection (ATA 30). For this prototype to go to production, the CDL should be included.

  • Line 512: about “Fix success rate”. I understand that you finally do not retain this rate. Nevertheless, as presented, this metric seems weird. Such a rate for a troubleshooting process makes sense, but applied to an OPC task (fig.5) I don’t think it is relevant (purpose of such a task is not to fix a failure). Maybe you can improve your wording to clarify the final expectation of this metric.

If I understand correctly you are referring to the Operational Test with a “fix success-ratio” of 35%. Indeed, for that option a term such as “resolve success rate” would have been a better fit. We have rephrased this part for clarification.

  • Lines 549: if I were a continuing airworthiness authority inspector, I would ask you to demonstrate that the app presents information fully equivalent to the up-to-date full set of maintenance data. Saying that I fully recognized that the risk of human errors when looking for data in the TechPub is maybe higher than the risk of not having correct information presented by the app… but regulation is what it is! Maybe you can improve a bit the text with the way you check the completeness and correctness of data. It is related to lines 579-583 and 588-593

For this specific case study we, researchers and technicians, manually verified that the dispatch options and task-support information shown in the prototype match what is found in the manuals or AirN@v. We have added a sentence after line 549 to clarify this. For a future version of the prototype an automated approach is preferred, we added this as a future research direction. W.r.t. being up-to-date, this can be achieved by either manually or automatically process the new maintenance manual(s) once an update has been released.

  • Lines 605-610 : latest aircraft generations such as A350 incorporate advanced AMCS or equivalent, with troubleshooting procedures. Maybe, if you have the appropriate knowledge that is probably in the scientific literature, you can position your prototype wrt those on-board systems (functionalities, performances, …)

From what we have seen at Airbus, their ACMS (AiRTHM) is an advanced aid for troubleshooting issues with health monitored parts and systems, and is marketed as a preventive service. The added value of this prototype is that can aid in troubleshooting of parts and systems, regardless of health monitoring, that unexpectedly have failed. We have addressed the relevance of decision support for unexpected failures in the introduction (lines 33-37).

Typo / Minor editing suggestions:

  • Line 15 : acronym DSS to be explained

Replaced with “decision support systems”

  • line 159 : "the" is doubled

Removed redundant “the”

  • line 513 : I guess you mean "from" instead of "form"

Correct. Already addressed through the previous comment about the fix success ratio.

We have indicated changes to the manuscript in red text.

Reviewer 3 Report

This work presents a new digital tool/system for managing maintenance of aircraft fleets. Relevant literature has been reviewed sufficiently, with a focus on the problem at hand. The proposed tool (and the underlying algorithm controlling its operation) is well presented in the paper, with the advantages and benefits obtained from its implementation clearly articulated and substantiated quantitatively. Overall, the paper is well written, though some of the coding aspects could be moved to an appendix (however it is something the authors may wish to consider doing and not a mandatory change). The case study methodological approach is instructive, which is another strong feature of this paper (has an applied character in addition to treating the technical problem).

Section 3.2, however, would need to be be broken down to a couple of subsections, as a way to improve readability.

Also, a Conclusions section (preferably using bullet points) is missing from this paper and should be added. This should discuss the limitations of this work, in conjunction with the already presented future directions.

Author Response

Dear reviewer,

Thank you for your time and effort in reviewing our submission. We appreciate your constructive feedback and have addressed your concerns as follows:

  • Section 3.2, however, would need to be broken down to a couple of subsections, as a way to improve readability.

We have added subsections to improve readability.

  • Also, a Conclusions section (preferably using bullet points) is missing from this paper and should be added. This should discuss the limitations of this work, in conjunction with the already presented future directions.

We added a conclusions section including a clearer layout. Limitations are addressed as part of this section (in 5.1, in conjunction with the future directions of the work as recommended).